# Caveats in Generating Medical Imaging Labels from Radiology Reports with Natural Language Processing

**Tobi Olatunji**                 TOBI@ENLITIC.COM
**Li Yao**                     LI@ENLITIC.COM
**Ben Covington**                 BEN@ENLITIC.COM
**Alexander Rhodes**               ALEX@ENLITIC.COM
**Anthony Upton**               ANTHONY@ENLITIC.COM

## Abstract

Acquiring high-quality annotations in medical imaging is usually a costly process. Automatic label extraction with natural language processing (NLP) has emerged as a promising workaround to bypass the need of expert annotation. Despite the convenience, the limitation of such an approximation has not been carefully examined and is not well understood. With a challenging set of 1,000 chest X-ray studies and their corresponding radiology reports, we show that there exists a surprisingly large discrepancy between what radiologists visually perceive and what they clinically report. Furthermore, with inherently flawed report as ground truth, the state-of-the-art medical NLP fails to produce high-fidelity labels.

## 1. Introduction

Modern machine leaning models in medical imaging requires large amount of high-quality training data. Unlike others, medical imaging labels are typically given in the format of free-text radiology reports that summarize findings and recommend follow-ups. In order to enable supervised learning, one extra step is needed to convert reports to discrete sets of labels – a process that may be automated by NLP. In fact, healthcare organizations and academic institutions who produce and possess medical data have begun to address this labeling issue by bootstrapping the image annotation process using Clinical NLP (Wang et al., 2017). This approach holds immense promise. For instance, publicly released datasets amount to hundreds of thousands of labelled studies (Irvin et al., 2019; Bustos et al., 2019), starting a new wave of machine learning models trained with richer examples.

Although convenient and highly scalable, automated processes typically come with their own limitations that directly influence the quality of the trained models, which in turn impact downstream patient outcome. Unlike other work that focuses on improving NLP model performance, we trace the problem of labeling noise and inconsistency to its source. We experimentally demonstrate the fundamental discrepancy between what radiologists perceive visually in imaging exams and what they choose to clinically report. We highlight the fact that most of the discrepancy is due to the concept of *clinically non-actionable findings*, which are often excluded in the deliverable of radiologists' workflow.

In particular, we have curated 1,000 chest X-ray (CXR) studies from a non-screening setting, the majority of which contain at least one abnormal finding based on their reports.

For each study, two sets of labels are generated by radiologists. One is based solely on viewing images (denoted as $y_{\text{img}}^{\text{rad}}$), and the other is based only on viewing reports (denoted as $y_{\text{txt}}^{\text{rad}}$). Preliminary analysis shows a high disagreement rate between the two. Furthermore, the state-of-art medical NLP (denoted as $y_{\text{txt}}^{\text{nlp}}$) produces further disagreement to $y_{\text{txt}}^{\text{rad}}$.

**Related work**  Raykar et al. (2009) addresses the problem of multiple annotators providing noisy labels. Irvin et al. (2019) evaluates against 1,000 manually labeled reports, then compares NLP extracted mentions, negations, and uncertainty labels to NIH Labeler (Peng et al., 2018). Hassanpour and Langlotz (2016) uses a set of 150 manually labeled reports as a validation set to compare performance between rule-based and machine learning methods. In their work on head CT reports (Zech et al., 2018), 1,004 manually labelled reports are used to evaluate NLP performance. A similar approach is taken by Sevenster et al. (2015). Unlike previous work, we aim to highlight the limitation of report-based annotation by using CXR studies, an imaging modality that is known to lack of specificity. Due to its challenging nature, even report labels from radiologists fall short, let alone those of NLP.

## 2. Experiments

**Data**  We curated a set of 1,000 chest X-ray studies, the majority of which have at least one finding based on the report text for review by two groups of expert radiologists. Group 1 reviewed images while Group 2 reviewed reports, indicating presence or absence of 4 categories of abnormalities– Global Abnormal ($\text{ABN}_1$), Cardiomegaly ($\text{ABN}_2$), Consolidation ($\text{ABN}_3$) and Foreign Body or Medical Device ($\text{ABN}_4$). We also compared automated label extraction on the reports using the state-of-the-art NLP (Irvin et al., 2019). Results are shown in Table 1 below.

Table 1: Overall performance benchmark of report labels against image labels, and NLP labels against report labels by abnormality.

| | | $\text{ABN}_1$ | $\text{ABN}_2$ | $\text{ABN}_3$ | $\text{ABN}_4$ |
|---|---|---|---|---|---|
| $y_{\text{img}}^{\text{rad}}$ | Num. of findings | 728 | 121 | 143 | 186 |
| $y_{\text{txt}}^{\text{rad}}$ | Num. of findings | 683 | 37 | 257 | 81 |
| | Precision w.r.t. $y_{\text{img}}^{\text{rad}}$ | 0.72 | 0.35 | 0.36 | 0.98 |
| | Recall w.r.t. $y_{\text{img}}^{\text{rad}}$ | 0.67 | 0.11 | 0.63 | 0.43 |
| | F1 w.r.t. $y_{\text{img}}^{\text{rad}}$ | 0.69 | 0.17 | 0.45 | 0.59 |
| $y_{\text{txt}}^{\text{nlp}}$ | Num. of findings | 726 | 240 | 291 | 131 |
| | Precision w.r.t. $y_{\text{txt}}^{\text{rad}}$ | 0.81 | 0.13 | 0.59 | 0.24 |
| | Recall w.r.t. $y_{\text{txt}}^{\text{rad}}$ | 0.86 | 0.86 | 0.67 | 0.38 |
| | F1 w.r.t. $y_{\text{txt}}^{\text{rad}}$ | 0.83 | 0.23 | 0.63 | 0.29 |

**Results**  Agreement (F1-score) between image and report findings is highest on the Global Abnormal label ($\text{ABN}_1$) (69%). Given this discrepancy, NLP reaches only 83% agreement

with report findings on the Global Abnormal label. Findings extracted by NLP therefore represent 55.6% of image findings (typically considered as the gold standard). The gap is apparently wider for other findings. We provide some insights on the failure cases below.

**Failure analysis**   Of the 1000 studies in this analysis, 239 reports (24%) were labeled as normal, disagreeing with image annotations. In 194 reports (20%) labeled abnormal, image review found no abnormality. Two expert radiologists reviewed selected cases from both sets. We summarize the insights in the analysis below.

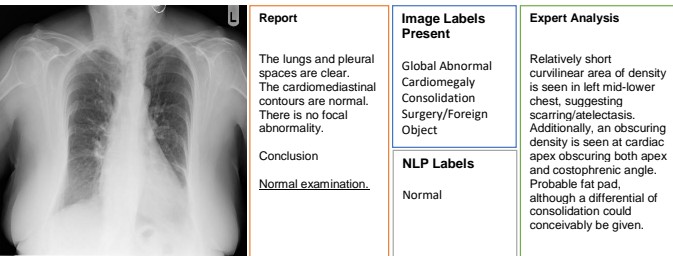

Figure 1: Technical issues and anatomic variations obscure findings

**Non-actionable findings**   In the overwhelming majority of disagreement, the reporting radiologist documents only findings relevant to the immediate clinical context (indication for ordering the study), and ignores non-actionable findings such as evidence of ongoing treatment (medical devices, leads, staples, catheters), unchanged findings (since previous study), age-related findings (in the elderly) such as spinal degenerative disease, spine arthritis, anterolateral osteophytes, aorta ectasia, calcifications, or curvature of the spine that do not contribute to primary pulmonary parenchymal pathology. The labeling radiologist however identifies such findings to provide consistent annotation for model training.

**Borderline or nuanced findings**   As seen in Figure 1, subtle findings demonstrate the low specificity of X-ray as a modality, leading to uncertainty and disagreement. Another clear example is *Borderline cardiomegaly* which results in the typical case of half-full vs half-empty where the labeling radiologist leans towards ignoring the finding, but the reporting radiologist might err on the side of caution, preferring to mention such findings with caveats.

**Anatomic variations**   Features such as barrel chest (pectus carinatum), skinny or obese patients, fat pads, nipple shadows, breast tissue density, superimposition of structures like ribs, cardiac shadow, create further ambiguity that result in suspicion of abnormality.

**Technical issues**   Other factors like patient positioning, inspiratory effort, image acquisition issues, clothing, nipple rings, medical devices, extrinsic or intrinsic foreign bodies influence the quality of report interpretation. They mask or exaggerate findings resulting in interpretation disagreement.

**Outright error**   In rare cases, reporting or labeling radiologists missed findings which the other picked up, or provided wrong interpretations to visual patterns. Such obvious errors (usually as a result of fatigue) strengthen the resolve to improve the performance of AI-assistsed radiology for patient care.

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
