# OpenReview forum: "Caveats in Generating Medical Imaging Labels from Radiology Reports with Natural Language Processing"
_MIDL.io/2019/Conference/Abstract — MIDL Abstract 2019_

### Official Review · AnonReviewer1 · 2019-04-29
**Important perspective on a recent research trend**

**Rating:** 3
**Confidence:** 2

**Review:**

Several groups have started to use NLP in order to automatically extract labels for supervised learning on medical imaging data from radiology reports. On a set of 1000 chest X-rays, the authors investigated to which extent radiologists arrive at the same labels based on the images and on their colleague's reports. They also evaluated to which extent labels extracted from the report via an NLP technique agree with those extracted from the same report by an expert. In both scenarios, the overlap is disappointingly small.

Even though these findings are not really specific to deep learning, and the call asks for works in which "deep learning is a key element", in my opinion, these findings should be relevant and interesting to the MIDL audience.

---

### Official Review · AnonReviewer2 · 2019-04-30
**Interesting study of noise in labels**

**Rating:** 3
**Confidence:** 2

**Review:**


Summary:

In this work, noise in annotations, observed as discrepancy between labels obtained from radiology reports and images are studied, using a curated dataset comprising 1000 chest X-ray scans and corresponding reports. Further, automatic text labels extracted using a recent NLP method is then shown to fare poorly as the underlying ground truth is noisy.

Comments:

+ Addresses an important issue about working with noisy labels. As new ML methods are trying to harvest on different types of labels (including weak labels), this is an important study highlighting discrepancies in annotations.
+ A well curated dataset, reasonably demonstrates the main points of this study
+ Insightful discussion of the results.

- Insufficient discussion of the NLP model that is used. While it is understandable that in an abstract version not sufficient details can be provided, a few key pointers about the NLP method can help understand if it is a reasonable approach to automatically extract the labels.

Other remarks:
I was wondering, if the non-actionable findings somehow be turned into labels that could still be used? For instance, just the images can be used to learn some vector embedding that can be then concatenated with the word2vec representation of the derived text labels.

---

### Decision · Program_Chairs · 2019-05-06
**Acceptance Decision**

Accept